# Confirmation of a hyperendemic focus of porcine cysticercosis in Northern Uganda: Prevalence and risk factor analysis

Nicholas Ngwili[1‡], Salaviriuse Ahimbisibwe[2,3‡], Max Korir[1], Stephen Bole[2,3], Clovice Kankya[3], Alfred Kinyera[4], Stanely Vusso Avudraga[5], Robert Saverio Okeny[6], Robert Kenny Okwera[7], Emily Ouma[8], Ian Dohoo[9], Lian F. Thomas [1,10]*

**1** Health Program, International Livestock Research Institute, Nairobi, Kenya, **2** Health Program, International Livestock Research Institute, Kampala, Uganda, **3** College of Veterinary Medicine, Animal Resources and Biosecurity, Makerere University, Kampala, Uganda, **4** District Veterinary Office, Kitgum District Local Government, Kitgum, Uganda, **5** District Veterinary Office, Lamwo District Local Government, Lamwo, Uganda, **6** District Veterinary Office, Pader District Local Government, Pader, Uganda, **7** District Veterinary Office, Agago District Local Government, Agago, Uganda, **8** People, Policies and InstitutionsProgram, International Livestock Research Institute, Kampala, Uganda, **9** Centre for Veterinary Epidemiological Research, University of Prince Edward Island, Charlottetown, Canada, **10** Royal (Dick) School of Veterinary Studies, University of Edinburgh, Edinburgh, United Kingdom

‡ These authors share first authorship on this work.
* lthomas8@ed.ac.uk

## Abstract

### Background

*Taenia solium* (*T. solium*), a neglected zoonotic tapeworm transmitted between humans and pigs, is a leading cause of acquired epilepsy in endemic areas where it is propagated by poor sanitation and pig husbandry practices. The World Health Organisation (WHO) NTD roadmap 2021–2030, recommended that targeted control interventions need to be initiated, and intensified in *T. solium* hyperendemic areas. Geospatial risk maps have identified Northern Uganda as a potential hyperendemic area. This study aimed to validate these findings and provide contextual evidence to support design and implementation of targeted interventions to control the parasite.

### Methods

A cross-sectional study was conducted in 2023 in four districts of northern Uganda. Blood samples were collected for serological analysis from 1049 pigs drawn from 714 households. Self-reported and observational data, and environmental variables from secondary sources were also collected. A subset of the seropositive pigs was dissected to confirm the presence of the parasite. The crude prevalence was adjusted for the test's sensitivity (Se = 0.867) and specificity (Sp = 0.947). Risk factors for seropositivity were evaluated using generalized mixed-effects models run in both R and Stata statistical software.

**Data availability statement:** All data for this study is openly accessible here: https://hdl.handle.net/20.500.11766.1/FK2/2FHY76

**Funding:** This research was made possible by funding from the German Federal Ministry for Economic Cooperation and Development (BMZ) through the One Health Research, Education and Out-reach Centre in Africa (OHRECA) led by ILRI (to LFT supporting NN, SA, SB and MK). We also acknowledge all CGIAR Fund Donors: https://www.cgiar.org/funders/ (supporting LFT, NN, SA, SB, MK & EO). The funders had no role in study design, data collection and analysis, decision to publish, or preparation of the manuscript.

**Competing interests:** The authors have declared that no competing interests exist.

## Results

The prevalence of porcine cysticercosis in this area was 17.4% (15.1– 19.7; 95% CI). Pig level predictors of infection were pigs that were eight months or older (odds ratio (OR)=1.88; $p = 0.001$), and non-local breeds of the pig (OR=1.7; $p = 0.01$). Household-level risk factors included the use of borehole water, (OR=6.39; $P = 0.001$), free-roaming pigs (OR=1.92; $p = 0.023$), whilst the presence of a toilet in the compound was protective (OR=0.64, $p = 0.05$).

## Conclusion

Our findings confirm that the study area is hyperendemic for *T. solium* infections, as the geospatial risk maps predicted. To achieve the targets laid out in the 2021–2030 WHO roadmap for control of NTDs, this region requires intensified targeted control interventions, preferably targeting both human and porcine hosts using the One Health approach.

## Author summary

*Taenia solium* (also known as the 'Pork tapeworm') is a neglected zoonotic parasite that causes infections in humans and pigs; taeniasis and cysticercosis in humans, and porcine cysticercosis in pigs. Ingesting the pork tapeworm eggs (normally due to open defecation by a tapeworm carrier) results in cysticercosis in both pigs and humans. A form of human cysticercosis where the pork tapeworm cysts lodge in the brain (neurocysticercosis) is the leading cause of adult-onset epilepsy in endemic areas. To reduce the burden of this debilitating disease, the World Health Organisation (WHO) included *T. solium* cysticercosis in the 2021–2030 Roadmap for the Control of Neglected Tropical Diseases. This roadmap recommends the implementation of control interventions within 'hyperendemic' areas. A cross-sectional study was conducted in this area covering four districts to validate the geospatial risk maps which identified hyperendemic areas within northern Uganda and to determine the prevalence and risk factors of porcine cysticercosis. The area was found to be hyperendemic for the pork tapeworm with a prevalence of 17.4% and pigs were more likely to be positive if they were free roaming, their household had no toilet, or the pigs were provided with borehole water.

## Introduction

Pig farming is a source of livelihood for 2.3 million households in Uganda [1] with most households relying on pig stock to meet short-term household needs such as medical bills, education fees and scholastic materials [2]. In the last 15 years, Uganda's pig production has more than doubled from 3.2 million pigs in 2009 to 6.7 million

in 2021, with Northern Uganda recording the fastest growth rate [1]. This rapid growth is aided by an increase in local and regional demand for pigs and pork [3,4]. Despite the immense opportunities the pork sector promises for Uganda's rural economy, public and private investments to modernize the industry remain limited. The pig value chain is unorganized and characterized by an inadequate extension system, a high burden of diseases, and the predominance of smallholder, sub-sistence, extensive systems [3,4]. Pig diseases are a continuous cause of economic losses, and are a risk to pig health, pig welfare and public health. Economic losses accrue from treatment costs, reduced pig productivity, death of pigs, the condemnation of pig carcasses at slaughter, and zoonotic disease transmission [4–6].

*Taenia solium* (pork tapeworm) is a neglected zoonotic parasitic worm, capable of causing three diseases: porcine cysticercosis (PCC) in pigs, and taeniasis and cysticercosis in humans. Humans are the definitive hosts for the mature tapeworm as well as the aberrant hosts in cases of human cysticercosis, while pigs are the intermediate hosts carrying metacestodes in the skeletal muscles and the brain [7]. Neurocysticercosis (NCC) occurring when the cysts are present within the brain, is a leading cause of adult-onset epilepsy in endemic areas [8]. NCC alone is estimated to cost Uganda 75 million USD in annual losses resulting from early death and disability [9]. Pork and infected pigs, with cysts, if found at meat inspection or identified by the traders, are condemned with no compensation for the animal owner. However some-times pigs identified as infected through tongue palpation at the point of purchase by traders are bought at a reduced price [4].

The transmission cycle of the pork tapeworm involves the shedding of tapeworm eggs by a human tapeworm carrier via the stool. Pigs get infected when they directly ingest the eggs or indirectly through contaminated feed and water due to environmental contamination with human faecal material containing eggs. Access to infective human faecal matter, contaminated feed, and water is a consequence of open defecation, a practice in over 2.3 million households across the Uganda [10]. In Uganda, the Northern region was highlighted by geospatial risk maps as a potential epidemiological hotspot for *T. solium* infections due to the overlap of three risk factors; high poverty rates, low latrine coverage and usage, and high pig density [11]. Earlier targets to eradicate or stop the transmission of NTDs including taeniasis and neurocys-ticercosis by 2020 were missed in most countries including Uganda. In 2021, the WHO set a new agenda, which Uganda ratified, to identify and implement control interventions in hyperendemic foci aimed at stopping the transmission of *T. solium* infections [12,13].

Although Uganda ratified WHO's NTD roadmap 2021–2030, achieving its targets remains unlikely due to: (i) the lack of national and subnational disease-specific control programs, (ii) the absence of an evidence-base confirming hyperen-demicity areas for most of the areas in Uganda (iii) a lack of prioritization of *T. solium* infections control within the national public health and One Health priorities [12] (iv) the lack of understanding of the local contexts that perpetuate the trans-mission and (v) lack of cross-sectoral coordination at local, regional, and national levels. These problems are not limited to the study area but are national issues that impede efforts to eradicate *T. solium* alongside other NTDs [12]. Factors perpetuating *T. solium* transmission such as open defecation, extensive pig production systems, and eating improperly cooked pork are still reported in the area [4,6,14,15].

The WHO Neglected Tropical Diseases (NTD) roadmap 2021–2030 recommends an 'intensified approach in hyperen-demic areas for *T. solium'*. While the term 'hyperendemic' is not further defined in the roadmap, targeted control in areas with high porcine prevalence is required to mitigate the risk of exposure to humans. As there is a lack of access to neuro-imaging within most endemic areas, NCC in humans is underreported and using the porcine host as a proxy for infection risk in humans has been previously recommended [16]. Our previous work identifying potential high-risk areas and review-ing the available *T. solium* prevalence data indicated that empirical data on the prevalence of *T. solium* was not available for Northern Uganda [11]. This study aimed to (i) verify our hypothesis that northern Uganda represents a hyperendemic focus where control interventions are required based on the risk mapping [4,11], (ii) identify additional risk factors that may support improved risk mapping exercises, and (iii) obtain robust prevalence data to inform future national and subnational intervention design and implementation.

## Materials and Methods

### Ethics Statement

This study was approved by the Institutional Research Ethics Committee (IREC-ILRI-IREC2022–66), IACUC (ILRI-IACUC2022–44), ILRI Institutional Biosafety Committee (ILRI-IBC2022–28) School of Veterinary Medicine and Animal Medicine, Research Ethics Committee- International Animal Care and Use Committee, Makerere University (SVAR_IACUC 136/2022), and research permit from the Uganda National Council for Science and Technology was obtained (A263ES).

### Study design

This cross-sectional study was conducted between April and October 2023 and included data collection using a pretested questionnaire administered to pig owners via open data kit (ODK) (https://getodk.org/) and blood sample collection from pigs.

### Study area selection

The initial selection of this area was based on an earlier spatial risk mapping for Uganda that highlighted Northern Uganda as potentially a hyperendemic foci for *T. solium* infections based on the congruence of the three risk factors namely poverty levels, pig density, and sanitation parameters [11]. Four districts of Northern Uganda; Kitgum, Lamwo, Agago, and Pader, where no prior studies have been conducted, and based on additional anecdotal reports of PCC infections from stakeholders in the area, were purposively selected [4,17]

### Description of the context of the study area

The Northern region of Uganda remains underdeveloped with high poverty levels, poor sanitation indicators, high youth unemployment and limited alternatives for household incomes [17]. Recent population statistics indicate that the average national population human density is 227 persons per sq km, but for the four districts, the average density is 64 people per sq km [18] (Table 1) and much of the area is covered by arable unused land dominated by vegetation and grassland. The total area of the four districts 16,383 square km with Lamwo being the largest district at 5,588 square km (Table 1).

### Sample size determination

The sample size for determining the prevalence of PCC was calculated using the formula $n = [Z^2P(1-P)]/d^2$ as defined by Thrusfield [19], where n is the required sample size; Z is the multiplier from a standard normal distribution (1.96) at a probability level of 0.05; P is the estimate of prevalence taken at 13.6% as reported by Alarakol et al [20] in Amuru and Gulu which border the districts under study; and d is the desired precision level (0.05). From the calculation, a sample size of

**Table 1. Human and pig population distribution across four districts.**

| District | Total HH[*a] | Livestock keeping HH[a] | Pig keeping HH[a] | Pig population[a] | Human population[b] | Area in square km | Population density[b](psq) |
|---|---|---|---|---|---|---|---|
| Agago | 39,106 | 33,223 | 8,073 | 20,337 | 213,799 | 3,503 | 89 |
| Kitgum | 70,569 | 59,745 | 11,411 | 41,506 | 239,386 | 3,998 | 55 |
| Lamwo | 41,525 | 36,257 | 4,822 | 16,166 | 310,424 | 5,588 | 38 |
| Pader | 60,541 | 53,367 | 10,887 | 28,080 | 240,133 | 3,294 | 72 |
| Total | 211,741 | 182,592 | 35,194 | 106,089 | 1,003,742 | 16,383 | 64 |

*HH – Household, psq = people per square kilometre which denotes unit for population density. The data are from the [a]National Livestock Census Report 2021 [1], and [b]National Housing and Population Census 2024 [18].

180 pigs per district was required for the study making a total of 720 pigs. We sampled 720 households from which a maximum of two eligible pigs were sampled. We confirmed that this sample size was sufficient for a risk factor analysis. We calculated the sample size for risk factor analysis using the formulae for 2-independent sample binomial comparison [21]. Absence of a latrine used as the risk factor of interest for this calculation with the prevalence of PCC in households with latrines assumed to be (12.7%) and those without (2.2%) as reported in Kungu et al., [15]. The target sample size to be able to find at least 95 households without toilets was 380 households based on 25% toilet coverage in the study districts indicating that our initial sample size was sufficient for risk factor analysis.

## Household selection

In each district, six sub-counties were purposively selected in consultation with district-level stakeholders, prioritizing those known to have high pig densities ensuring representation of different geographical locations within the district and excluding those with ongoing security concerns. The lists of villages per sub-county were obtained from each Local Government and 6 villages were randomly selected in Microsoft Excel. Extension workers working with village chairpersons provided a list of all pig-rearing households in each village. Five pig-keeping households were randomly chosen using a mobile-based Randomizer App [22]. If the household did not consent or had no eligible pigs or the pigs could not be captured, the remaining households underwent random selection by the field team to select a replacement. If we did not get 5 qualifying households, additional households were picked from a nearby village. In each village, the village chairperson sensitized the members a day before our visit.

## Pig selection

In each household, a maximum of two eligible pigs were selected for sampling. A pig was eligible to be sampled if three months of age or older, was not pregnant, and was not physically emaciated to avoid complications during blood sampling. Sampling was convenient, with the most readily captured eligible pigs being those sampled. if no eligible pigs were present an additional household was recruited to replace the household.

## Data collection

The household-level questionnaire underwent pretesting in 2 villages in Kitgum district not included in the target sites. The questionnaire was pretested through cognitive interviewing to test the understanding and flow of the questions, and adjustments were made and a final version uploaded on ODK. It was then administered by a trained enumerators to either the household head, spouse, or a permanent household individual member knowledgeable about the management of the pig enterprise in the household. Data were collected on self-reported and observational variables namely pig housing, sources of water and feed, self-reported family deworming, pork slaughter and pork consumption patterns, observed presence of toilet and signs of use, and knowledge of *T. solium* infections and control (S1 Table).

Several environmental variables were also integrated into the dataset, including the distance from the household to the nearest river, road, and health centre using the distance to the nearest hub (points) in QGIS (S2 Table). Data on the average of rainfall, altitude, normalised difference vegetation index (NDVI), and slope means were extracted for a buffer of 300m for each household, assumed to be the average roaming limit of pigs in the area [23].

## Sample collection and analysis

Pigs were restrained using a pig snare or held in dorsal recumbency if under approximately 10 kg. Blood samples were taken from the anterior vena cava on the pig's right side into a barcoded BD Vacutainer 10-ml tubes. The vacutainer was immediately placed into a cool box and later transported to the district veterinary laboratory at the end of the day, where they were centrifuged at 3000 rpm for 10 min at room temperature. Sera were then separated into two aliquots in 2-ml

cryovials labelled with a unique barcode and stored at −20 °C until they were transported on dry ice to the Central Diagnostic Laboratory, Makerere University for laboratory analysis at the end of the data collection exercise.

ApDia (ApDia group, Belgium) Antigen Enzyme-Linked Immunosorbent Assay (AgELISA) was used based on the manufacturer's instruction to determine the seropositivity of each pig sample [24]. The tests were run in duplicate microtiter wells and the optical density (OD) of each sample was measured at 450 nm with a reference wavelength of 630 using a microplate reader (Biochrom, Cambridge – CB4 0FJ, England) with the cut-off value calculated per plate as the mean OD of the negative control multiplied by 3.5. For each well, the Antigen index was calculated by dividing the mean OD value by the cut-off. A sample was classified as negative if the Ag index was ≤ 0.8, positive if ≥ 1.3 and suspect if between 0.8 and 1.3.

## Carcass dissection

Twenty (20) out of the 204 seropositive pigs were purchased and dissected to confirm the presence of *T. solium* cysts. The pigs were located at the farmer's home and if the farmer consented, the pigs were bought at the prevailing market price and transported to a local butchery premise. The pigs were then euthanised by exsanguination through cutting of the major vessels of the neck. The carcass dissection was conducted according to the protocol suggested by Lightowlers and others [25]. *T solium* metacestodes were recorded as viable or non-viable using the criterion suggested by Sah et al [26]. This was aimed at confirming the presence of *T. solium* cysticercosis in pigs considering the cross-reactivity of the ELISA test used with *T. hydatigena. T. hydatigena* were identified on gross pathology as large (>1 cm) thin-walled fluid filled cysts with long slender necks at the attachment with liver capsule or other peritoneal surfaces rather than within the musculature.

## Data analysis

Descriptive and summary statistics for all the variables were computed, and p-values determined using Fisher's exact test [27] in R [28]. The overall and district-level prevalence of *T. solium* in the districts was determined following adjustment for test specificity (0.947) and sensitivity (0.867) [29].

A causal diagram, or directed acyclic graph (DAG), was generated online using DAGitty (https://www.dagitty.net/) to identify the relationships among the potential predictors, possible confounders, likely interactions, and random effects (Fig 1). Risk factors included pig husbandry practices (e.g. free roaming), availability of a latrine in the household compound, pig demographics (e.g. breed, age), selected interactions among factors which were deemed to be biologically plausible and contextual effects (all risk factors explored are given in S1 Table).

First, the variables were tested for unconditional association, with some variables dropped at this level if they were considered to have a weak biological plausibility. Univariable analyses between pig characteristics, household respondent characteristics, pig management practices, household sanitation, pork consumption, environmental variables, and the outcome of interest (PCC seropositivity) were computed using a random effect logistic regression model with household nested within the village as random effects to account for the clustering of infections within the household and across the village as the pigs roam in the village with districts were included as fixed effects.

Based on the DAG (Fig 1), a multivariable model was built, with the main variables of interest being the presence of a toilet in a household, free-roaming pigs, watering with borehole water, slaughtering pigs at home and eating pork out. The final model variables were selected for consideration based on their statistical significance (P < 0.15) and the likelihood of being confounders with the final variables reported being those with (P < 0.05). Although environmental variables were computed at the household level, they were highly consistent across all households in a village, so they were treated as village-level variables. With the limited number of villages and hence limited power to detect effects at this level, environmental variables were evaluated by adding them one at a time to model with important pig and household level variables. Those with P<=0.15 were retained for multivariable modelling.

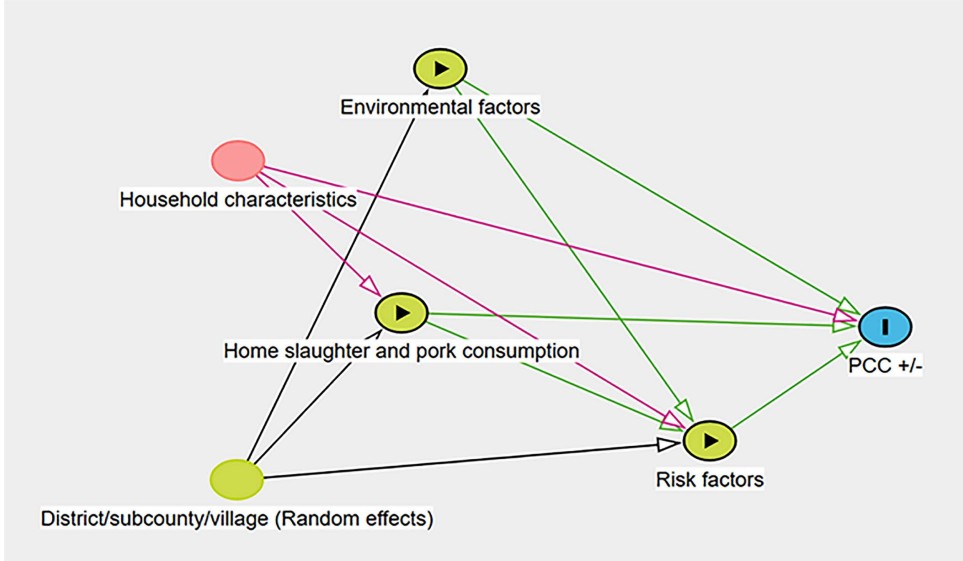

**Fig 1. Directed Acyclic Diagram (DAG) showing the relationship between factors potentially affecting PCC.** The causal diagram lays out the hypothetical causal relationships between variables with the direction of arrows indicating the possible causal relationship between different variables. For many risk factors, it would be difficult to assign a definitive causal sequence (e.g., does roaming influence pig deworming or vice versa) so they were placed into a single node. The outcome variable is PCC +/- with exposure variables being to the left of the outcome variable (e.g.,: Environmental factors). The variable to the left of the exposure variables are potential confounders and are included in the model. The green arrows represent causal paths while the pink arrows represent a biasing path.

Interactions between the following pairs of variables were selected for evaluation based on their biological plausibility (determined a priori): eating pork out of the household/family report deworming, eating pork out of the household/presence of a toilet in the compound, and pig kept free-roaming/family report deworming. Contextual effects for all risk factors identified as significant at the pig level were computed as the average of the risk factors at the village level (e.g., both pig roaming and the proportion of pigs that roam in the village) were considered as risk factors.

## KMEANS clustering analysis

In our analysis, we employed K-means clustering to investigate the spatial distribution of *Taenia solium* cysticercosis by analyzing village locations and cysticercosis infection statuses. To identify the optimal number of clusters, we utilized the Elbow and Gap statistic methods (S1 Text). These methods helped determine the most appropriate grouping of villages based on their cysticercosis infection status and geographical location.

To evaluate the effectiveness of the clustering, we analyzed several metrics including the Within-Cluster Sum of Squares (WCSS), the Between-Cluster Sum of Squares (BCSS), and the Total Sum of Squares (TSS). The WCSS measures the sum of squared distances between data points and their respective cluster centroids, with lower values indicating tighter, more cohesive clusters. On the other hand, the BCSS, which captures the sum of squares between the centroids of distinct clusters and the overall data centroid, provides insight into the separation quality between clusters; higher values indicate better separation. The TSS represents the total variation within the dataset and is decomposed into WCSS and BCSS to assess cluster quality.

For a comprehensive evaluation of clustering quality, we visualized the silhouette scores, which measure how similar an object is to its own cluster compared to other clusters, providing a clear visual representation of clustering efficacy. Additionally, Jaccard similarity values were computed using a bootstrapping approach to further validate cluster reliability and ensure robustness and accuracy in the results obtained.

## Development of the final model

The district was included in all multivariable models as a fixed effect to account for that level of the hierarchy. Village and household were included as random effects. Respondent characteristics were included in the model to account for potential confounding. None of the interactions, nor the contextual effects were significant and hence not included in the final model. Other predictors were retained if P<.05 or P<0.15 for environmental variables.

Model building was carried out in R, but final models were fit in Stata to take advantage of the ability to use adaptive quadrature estimation of the likelihood function [30]. Estimates of household and village-level variances were obtained from this model. The intraclass correlation coefficients (ICCs) at these levels were computed assuming the lowest level variance was 3.29 [31]. Random effects (at the household and village levels) from the final model were computed and evaluated for normality, homoscedasticity and the presence of extreme outliers.

## Results

### Descriptive statistics

A total of 1049 pigs were sampled from 714 households from 165 villages across 24 sub-counties in 4 districts (S3 Table). The mean and median age of sampled pigs was 12.07 and 8 months respectively, with actual ages ranging from 3 to 120 months. More female pigs (62.8%) were sampled than male pigs (37.2%). Older pigs were more likely to be female than males and this difference was significant (p<0.01) (Tables 2 and S5 Table).

In more than half of the households the household head was interviewed with majority being male (Table 3). Majority of the respondents had primary level of education and below with over half aged below 39 years of age.

Various pig husbandry practices were reported that may pose an infection risk for pigs (S4 Table). Allowing pigs to roam was a common practice as reported by 83.1% of the households. Overall, there was no difference in confinement of younger pigs compared to those above 8 months. The mean and median of the pig herd size were 4.15 and 2 respectively, the majority (63.2%) of households had less than 3 pigs and the herd sizes ranged from 1-46 pigs. Herd sizes varied by district (p<0.01). Various feed sources were mentioned as frequently used by the pig producers, in addition to free roaming their pigs. In the study area, the most common feeds were fodder and crop residues (81.4%), commercial feeds (50.3%), brewer's waste (23.5%), and swill (19.2%).

Most of the households reported that boreholes were the most common source of water for watering the pigs (92%), ranging from 86.5% to 95% of households in the various districts (p=0.003). Other reported water sources included rainwater and surface water. Overall, only 29.4% of respondents reported ever deworming pigs, but for specific districts,

**Table 2. Pig level characteristics.**

| Pig level characteristic | Agago (n=270) | Kitgum (n=251) | Lamwo (n=265) | Pader (n=263) | Total (N=1049) |
|---|---|---|---|---|---|
| **Pig sex** | | | | | |
| Male | 43.3 | 34.7 | 35.8 | 34.6 | 37.2 |
| Female | 56.7 | 65.3 | 64.2 | 65.4 | 62.8 |
| **Pig age** | | | | | |
| Less than 8 months | 75.5 | 47.1 | 57.7 | 72.6 | 63.7 |
| 8 months or older | 24.5 | 52.9 | 42.3 | 27.4 | 36.3 |
| **Breed** | | | | | |
| Local | 73.3 | 26.3 | 73.6 | 66.5 | 60.4 |
| Exotic/crossbreed | 26.7 | 73.7 | 26.4 | 33.5 | 39.6 |

**\*Numbers represent percentages**

**Table 3. Demographic characteristics of the respondent household.**

| Demographic characteristic | Agago (n = 179) | Kitgum (n = 180) | Lamwo (n = 178) | Pader (n = 177) | Total (N = 714) |
|---|---|---|---|---|---|
| **Respondent** | | | | | |
| Household head | 55.3 | 56.1 | 52.2 | 57.6 | 55.3 |
| Delegate | 44.7 | 43.9 | 47.8 | 42.4 | 44.7 |
| **Gender of the respondent** | | | | | |
| Male | 50.8 | 60.6 | 60.7 | 55.4 | 56.9 |
| Female | 49.2 | 39.4 | 39.3 | 44.6 | 43.1 |
| **Age of the respondent** | | | | | |
| 39 years or below | 55.3 | 50.6 | 56.7 | 56.5 | 54.8 |
| Above 39 | 44.7 | 49.4 | 43.3 | 43.5 | 45.2 |
| **Education** | | | | | |
| Primary education or below | 82.1 | 68.3 | 59.0 | 87.0 | 74.1 |
| Above primary education | 17.9 | 31.7 | 41.0 | 13.0 | 25.9 |

**\*Numbers represent percentages**

the deworming rates ranged from 22.9% to 37.8% (p = 0.02). Most households (94%) reported that at least one of the household members consumed pork (Table 4). Of these, 564 sometimes consumed pork outside the home at pork joints, restaurants, roadside roasteries, or village events. Some households (20.2%) reported that they sometimes slaughter pigs at home.

Overall, 24.1% of households did not have a toilet or pit latrine indicating open defecation, although some of them reportedly use the neighbour's latrine. There were differences in toilet coverage across the 4 districts ranging from 76.5% in Lamwo down to 62.0% in Kitgum, and these differences were statistically significant across the districts (p < 0.01). However, some of the households (n = 76) that had toilets were also observed to practice other forms of open defecation such as not using the toilet or disposing of the children's faces in the bush. Deworming the family was reportedly undertaken 'sometimes' in 55% of the households while the remainder reported that they never dewormed, with differences (p < 0.05) across the districts.

**Table 4. Sanitation, deworming and pork consumption practices.**

| Sanitation and pork consumption practice | Agago (179) | Kitgum (n = 180) | Lamwo (n = 178) | Pader (n = 177) | Total (N = 714) |
|---|---|---|---|---|---|
| **Eat pork home** | | | | | |
| No | 2.8 | 8.3 | 6.2 | 6.8 | 6.0 |
| Yes | 97.2 | 91.7 | 93.8 | 93.2 | 94.0 |
| **Eat pork out** | | | | | |
| No | 15.1 | 26.1 | 28.7 | 16.4 | 21.0 |
| Yes | 84.9 | 73.9 | 71.3 | 83.6 | 79.0 |
| **Slaughter pigs at home** | | | | | |
| No | 90.5 | 71.1 | 74.2 | 83.6 | 79.8 |
| Yes | 9.5 | 28.9 | 25.8 | 16.4 | 20.2 |
| **Toilet in the compound seen** | | | | | |
| No | 38.0 | 16.7 | 13.5 | 28.2 | 24.1 |
| Yes | 62.0 | 83.3 | 86.5 | 71.8 | 75.9 |

**\*Numbers represent percentages**

## Prevalence and distribution of PCC in the study area

The prevalence of PCC was calculated at the individual pig level, household, and village level (Table 5). The apparent pig level prevalence was 19.5% (17.1-22.0, 95% CI). The overall prevalence adjusted for test specificity and sensitivity was 17.4% (15.1-19.7, 95% CI). The pig level prevalence differed across the four districts, and the difference was significant (p = 0.017, Fisher's Exact Test). At the household level, out of 714 households sampled, 181 were found to have at least one positive pig translating to a prevalence of 25.4% (22.2 - 28.7, 95% CI). There was no statistically significant difference in the observed prevalence (p = 0.11) across the four districts. Overall, 104 (63.4%; 55.5 - 70.8, 95% CI) out of 165 villages had at least one positive case of PCC, and although there were differences across the four districts, the differences were not significant (p = 0.08). Porcine cysticercosis cases were distributed across the study area, and analysis of K-means clustering revealed an extremely low within-cluster sum of squares values (3.65e-26) and a high between-cluster sum of squares (1065) (S1 Text), Fig 2.

The map was produced using open source QGIS software version 3.34 (https://www.qgis.org). The datasets for Uganda were freely obtained from DIVA-GIS (https://diva-gis.org/data.html). The shapefile for African boundaries were also freely downloaded from ArcGIS hub (https://hub.arcgis.com/datasets/africa::africa-countries/about) and dissolved in QGIS.

The ICC for two pigs within the same household was 0.29 while for two pigs within the same village (but different households) was 0.11. Random effects and residuals were computed and evaluated graphically. Residuals at the lowest level of a logistic model are not useful as they can take on only two possible values [25]. The same problem extended to the pig-level random effects as most households contributed only 1 pig to the data set. The village-level random effects were approximately normal and homoscedastic (S1 and S2 Fig). No extreme outliers were detected.

## Carcass dissection

Twenty seropositive pigs were dissected as previously described and cysts were enumerated. Out of the 20, 12 had viable *T. solium* cysts in skeletal muscles and the brain, four had degenerated cysts or lesions suggesting that the cysts had resolved, while five had *T. hydatigena* cysts: 4 with *T. hydatigena* only and 1 with a mixed *T. solium – T. hydatigena* infection. In one of the carcasses, no lesions of viable or resolved cysts were seen. The number of viable *T. solium* cysts ranged from 19 to more than 10,000 per pig, with nine out of the 12 pigs having at least 1000 cysts in the carcass. For *T. hydatigena*, four pigs had one cyst each while the fifth pig had two cysts, and these were recovered from the mesenteries, spleen, or liver.

## Risk factors for PCC seropositivity

Each of the variables were unconditionally tested for association (S5 Table). At the univariable modelling, thirteen factors were tested for association with PCC seropositivity after being selected based on their biological plausibility with the help of DAGs (Table 6). At the individual pig level, there were significant differences in prevalence for pigs that are eight months or older (p < 0.01) and the non-local pig breed (p < 0.01). The statistically significant household-level predictors were allowing pigs to roam, having no toilet in the household, and the use of borehole water for watering pigs (p < 0.05). Variables with p-values below 0.15, variables that were components of interaction terms of specific interest, and

**Table 5. Prevalence at pig, household, and village level.**

| District | Agago | Kitgum | Lamwo | Pader | Overall |
|---|---|---|---|---|---|
| Pig level (95% CI) | 12.7 (8.8–16.7) | 25.8 (20.4–31.2) | 15.3 (11.0–19.6) | 16.2 (11.8–20.6) | 17.4 (15.1–19.7) |
| Household Prevalence (95% CI) | 20.7 (14.7–26.6) | 31.7 (24.9–38.5) | 24.2 (17.9–30.5) | 24.9 (18.5–31.2) | 25.4 (22.2–28.5) |
| Village level (95% CI) | 61.5 (46.3–76.8) | 79.0 (66.0–92.0) | 52.1 (38.0–66.2) | 64.1 (49.1–79.2) | 63.4 (55.5–70.8) |

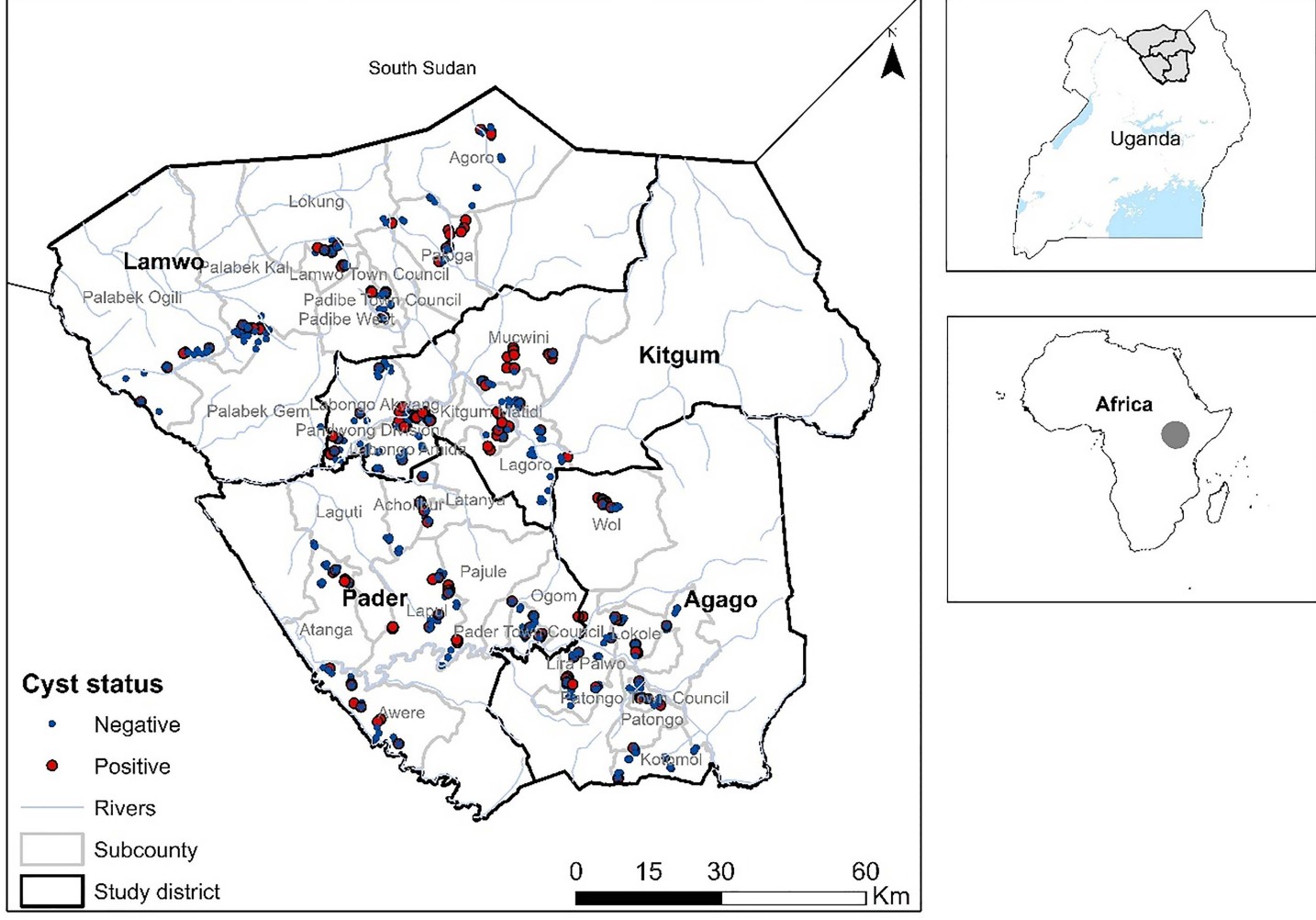

**Fig 2. Distribution of cases across the study area.**

household characteristics deemed to be potential confounders were retained for multivariable analysis. Pig level characteristics were intervening variables and were excluded from the model.

Environmental variables were also unconditionally explored but none were found to have a statistically significant association (P < 0.05) with PCC (S6 Table). Nevertheless, two variables (mean NDVI and mean slope) were retained in the final model because they had P ≤ 0.15 and there was very little power to detect meaningful effects of village-level factors.

### Final model

The results of the final model have been presented below (Table 7).

### Discussion

This study contributes to the growing evidence that *T. solium* is endemic in Uganda and hyperendemic in certain regions [15,32–36]. In addition, this study validates the potential use of geospatial modelling as a tool to predict hyperendemic areas where prevalence studies can be conducted to confirm endemicity as was done prior to the current study [13].

**Table 6. Risk factors associated with pig-level seropositivity of Porcine cysticercosis based on univariable logistic regression with a random effect for households nested with the villages.**

| Variable/Category | Levels | Prev/% | Odds ratio (95% CI) | P value | Overall P-value |
|---|---|---|---|---|---|
| District | Kitgum | 26.3 | 1 (ref) | | 0.055 |
| | Lamwo | 17.7 | 0.54 (0.30–0.98) | 0.044 | |
| | Agago | 15.6 | 0.44 (0.24–0.82) | 0.01 | |
| | Pader | 18.6 | 0.56 (0.31–1.03) | 0.062 | |
| Respondent | Household head | 20.1 | 1 (ref) | | 0.829 |
| | Delegate | 18.6 | 0.96(0.65–1.41) | 0.829 | |
| Gender | Male | 20.1 | 1 (ref) | | |
| | Female | 18.6 | 0.96 (0.65–1.41) | 0.829 | 0.829 |
| Education level | Educ 0 | 18.6 | 1 ref | | 0.614 |
| | Educ 1 | 22.2 | 1.24 (0.79–1.93) | 0.351 | |
| | Educ 2 | 20.8 | 1.19 (0.63–2.28) | 0.591 | |
| Age of the respondent | Less than 39 | 20.7 | 1 (ref) | | 0.332 |
| | 39 years or older | 17.9 | 0.82 (0.56–1.22) | 0.332 | |
| Sex of the Pig | Male | 16.7 | 1 (ref) | | 0.101 |
| | Female | 21.1 | 1.37 (0.94–2.00) | 0.101 | |
| Age of the pig | Less than 8 months | 16.1 | 1 (ref) | | 0.001 |
| | 8 months or more | 25 | 1.88 (1.28 – 2.74) | 0.001 | |
| Breed of the pig | Local | 16.7 | 1 (ref) | | 0.01 |
| | Non-local | 23.6 | 1.70 (1.13–2.54) | 0.01 | |
| Free-roaming pigs | Total confinement | 13.2 | 1 (ref) | | 0.023 |
| | Any form of free-roaming | 20.8 | 1.92 (1.10–3.37) | 0.023 | |
| Borehole water | No | 4.6 | 1 (ref) | | 0.004 |
| | Yes | 26.2 | 6.39 (2.10–19.40) | 0.004 | |
| Home slaughter | No | 18.5 | 1 (ref) | | 0.343 |
| | Yes | 23.3 | 1.26 (0.78–2.03) | 0.343 | |
| Eat pork out | No | 17.2 | 1 (ref) | | 0.365 |
| | Yes | 20.0 | 1.26 (0.77–2.06) | 0.365 | |
| Toilet in the compound | No | 22.1 | 1 (ref) | | 0.051 |
| | Yes | 18.0 | 0.64 (0.41–1.00) | 0.051 | |

Geospatial models have been used in East Africa to model future disease scenarios using past outbreak data as in the case of Rift Valley fever [37], malaria [38], and other NTDs [39,40], although few were reported to have been validated using empirical data from the field. This study has been able to validate geospatial risk mapping as a potential tool to identify likely hyperendemic areas for targeted control interventions in line with recommendations of the NTD roadmap 2021–2030 [41]. Based on the findings of this study our modelling approach has proven to be accurate and useful; however, more studies are needed to cross-validate the models in more diverse contexts including other endemic areas of Uganda. Identification of areas where interventions should be initiated or intensified can help achieve the goals set in the NTD roadmap.

The reported prevalence indicates that the area is hyperendemic (17.4%) for PCC and is higher than previously reported in the neighbouring districts of Gulu and Amuru (13.6%) using lingual palpation [20], and Moyo and Lira districts (10.4%) using Antigen ELISA [42]. The herd-level prevalence reported in our study (25.4%) is also higher than what has been reported in Lira (11.4%) and Moyo (13.7%) [43]. With 17.4% PCC infection and a quarter of households having at least one case, the risk for humans to acquire taeniasis through eating parasitized pork is high. This risk in the study area

**Table 7. Results of the final model.**

| Variable | Coefficient | Std. error. | z | P > \|z\| | [95% conf. interval] | |
|---|---|---|---|---|---|---|
| District | | | | | | |
| 2 | −.340052 | .3327045 | −1.02 | 0.307 | −.9921408 | .3120368 |
| 3 | −.6690454 | .3466768 | −1.93 | 0.054 | −1.348519 | .0104287 |
| 4 | −.3748554 | .3364348 | −1.11 | 0.265 | −1.034255 | .2845447 |
| Gender | −.2044441 | .26863 | −0.76 | 0.447 | −.7309491 | .322061 |
| Age of the respondent | .0432222 | .2242625 | 0.19 | 0.847 | −.3963242 | .4827687 |
| Education level | .2425577 | .276809 | 0.88 | 0.381 | −.299978 | .7850934 |
| | .0854999 | .178789 | 0.48 | 0.632 | −.2649202 | .4359199 |
| Free roaming pigs | .6248926 | .3059899 | 2.04 | 0.041 | .0251634 | 1.224622 |
| Borehole water | 1.8395 | .6027285 | 3.05 | 0.002 | .658174 | 3.020826 |
| Toilet in the compound | −.5053616 | .2420317 | −2.09 | 0.037 | −.9797351 | −.0309881 |
| Age of the pig | .766306 | .2065319 | 3.71 | 0.000 | .3615109 | 1.171101 |
| Mean NDVI | 1.053084 | .7073292 | 1.49 | 0.137 | −.3332555 | 2.439424 |
| Slope mean | −.4853216 | .3374563 | −1.44 | 0.150 | −1.146724 | .1760807 |
| Constant | −4.104232 | .988929 | −4.15 | 0.000 | −6.042497 | −2.165967 |
| Village | | | | | | |
| var(_cons) | .4859979 | .2612849 | | | .1694364 | 1.393998 |
| village>household | | | | | | |
| var(_cons) | .850268 | .5902675 | | | .2180923 | 3.314907 |

is exacerbated by inadequate pork inspection, food insecurity, lack of understanding of the risks, and reported inadequate cooking of pork [4]. The high pig and herd-level prevalence also indicates widespread pockets of environmental contamination with the eggs of the pork tapeworm presenting a cysticercosis risk for humans. Control interventions using the One Health approach are therefore necessary to simultaneously address the burden in pigs, humans, and the environment.

The spatial distribution across the study areas indicates a wide distribution of *T. solium* infections. Out of 164 villages, 104 had at least one positive case. The ICC estimates showed that *T. solium* infections clustered quite strongly within households (ICC = 0.29) and moderately within villages (ICC = 0.11). This supports the decision to sample a few pigs within a household to maximize the number of households and villages that could be sampled. The high between-cluster sum of squares indicates that the clusters are well-differentiated from each other. Notably, there is a significant difference in village values between the two clusters, suggesting a spatial or regional differentiation that might represent distinct geographic zones or socioeconomic contexts. This highlights the need for additional work to determine these key drivers to inform control efforts which consider these geographic distinctions. Both the ICC and K-means reveal hierarchical geospatial clustering of PCC cases. This clustering of cases has been reported in Tanzania [44] and Peru [45] and may be an indicator of environmental contamination resulting from an open-defecating human taeniasis carrier. This also implies that in the design of interventions, diligence must be taken to identify and address both household-level and village-level factors that facilitate the hierarchical infection risks. These results also support the feasibility of ring treatment strategies which an intervention ring is opened around an identified taeniasis case as has been done in Peru [46].

Although many factors and interaction terms expected to influence the seropositivity of pigs with PCC were investigated, only pig age and three husbandry factors were significantly associated at P < 0.05. This does not indicate that other factors investigated did not play a role, only that perhaps our sample size (n = 1049) was not able to detect those effects. Nevertheless, our findings provide a strong basis for a need to initiate control interventions in the study area in accordance with the NTD roadmap. The presence of hyperendemic areas, where the ongoing transmission is un-interrupted,

such as this one in Northern Uganda, presents a challenge for national and global efforts aimed at eradicating the parasite.

In this study, pigs aged 8 months or older had higher odds of infection compared to younger pigs. This may be because the random chance of a pig ingesting infective material increases with time, the older the pig the higher the chances. Also, older pigs will tend to roam further away from home increasing the area over which they can pick faeces [45,47]. There was evidence that the odds of infection were higher (OR=1.5) for non-local breeds than for the local breeds, statistically significant (P<0.05). Although the cause for this remains speculative, it has also been observed in other areas where the parasite is endemic including Uganda [15,35] and South Africa [48].

Similar to this study, other studies in other parts of Uganda also found pig roaming to be a risk factor for PCC [14,49]. We observed that pigs roamed far from their households within the village and may cross to nearby villages. In the study area, it has been reported that pigs are deliberately left to roam to look for feed and, in some instances, to clean the environment of dirt including human faecal matter [4]. Extensive pig production systems are highly prevalent even in other areas in the region [23,50]. One of the implications of pig roaming is that any open-defecating household in the roaming zone of the pig can be a potential source of infection to the roaming pig [45]. The wide distribution of PCC cases also implies that taeniasis cases are widespread, necessitating region-wide human and environmental-focused interventions to disrupt the transmission substantially and to reduce the burden in all three One Health domains.

Households with a toilet were found to have lower odds of *Taenia spp* infection. The lack of a toilet (open defecation) is frequently reported as a risk factor for PCC in endemic areas [51]. The rate of open defecation is lower than what was reported by Alarakol and colleagues [20] but is within national averages reported by the Uganda Bureau of Statistics [1]. Whereas, the absence of a toilet can be an indicator of open defecation, even households with latrines can have some members open defecating [49,52], especially children, resulting in environmental contamination. This is because, in most parts of Uganda, open defecation is not just an indicator of the absence of the toilet in a household but rather a socioeconomic and cultural issue, exacerbated by a lack of knowledge of the health risks associated with open defecation [47]. Open defecation is a necessary factor in the transmission cycle of the pork tapeworm for exposing humans and pigs to viable parasite eggs in faeces. In other areas, the use of human faeces as agricultural manure has been indicated to result in the transmission of the pork tapeworm. Refined attempts to quantify faecal contamination of the environment beyond the presence/absence of a latrine in the study area should be further investigated.

Borehole water was identified as a significant predictor in this study even though only 8% of households used other water sources. Although we suspect that boreholes might be an indicator of other village-level factors that we may have missed or possibly is a confounder variable; as households that use borehole water are as likely to be those with low-income levels as compared to those who use tap water. Contamination of boreholes by nearby pit latrines has been indicated to result in bacterial waterborne illnesses [53,54], but no study yet has demonstrated helminth egg contamination in boreholes presenting an opportunity for further studies.

Further studies focusing on the distribution of taeniasis and environmental contamination with *T. solium* eggs would be of great importance in understanding the observed patterns in this study. In Tanzania, a study on environmental contamination of the soil with *T. solium* eggs established a prevalence of 3.1% using droplet digital Polymerase Chain Reaction (ddPCR), although a major limitation could have been the targeting of the spots where the soil samples were picked [55]. Additionally, the presence of eggs and survival of the eggs may be influenced by environmental and climatic factors including temperature, humidity and vegetation cover [55]. In the current study, we attempted to incorporate environmental factors to help explain the trend of the infection, but no significant associations were found possibly because these factors exert influence at the village level so the power to detect effects was limited by the number of villages in the study. Nevertheless, this is the first study in the Ugandan context to explore environmental and socio-economic factors, in addition to exploring village-level contextual effects that might increase the likelihood of pigs becoming infected with PCC and how they may affect control interventions.

Acknowledging the performance issues with the currently available serological tests, our study carried out carcass dissection on a subset of the seropositive pigs to confirm the presence of *T. solium* cysticercosis in pigs considering the potential cross-reaction with *Taenia hydatigena* and identified many infected pigs with massive infections. Fine dissection of pigs is resource intensive and the removal of large numbers of pigs has the potential to disrupt the pork market in a region therefore we only carried out a limited dissection effort. Through this limited effort, however, we were able to confirm that both *T. solium* and *T. hydatigena* were present confirming the endemicity of both parasites. According to our knowledge, we are the first to report the coinfection of a pig with *T. solium* and *T. hydatigena* in Uganda, although this has been reported before at the Thai-Myanmar border [56]. This coinfection is contrary to the long-held hypothesis that density-dependent interspecies immune interactions prevent the concurrence of *Taenia spp* in pigs [57]. Future larger studies using carcass dissection would shed light on the frequency of *Taenia spp* coinfection in pigs. One of the carcasses had barely visible cysts indicating a likelihood of missing some cases by carcass dissection, although it remains the gold standard. Perhaps, in some cases, the use of a magnifying glass may be a necessary addition to the standard full carcass dissection suggested by Lightowlers *et al.* [25].

As observed above, there is co-endemicity of *T. solium* and *T. hydatigena* which are known to cross-react on the ApDia antigen cysticercosis test implying that indeed some of the positive cases attributed to *T. solium* were infections of *T. hydatigena*. The applicability of full carcass dissection in a large study such as this one is complicated as the team would not be able to dissect all the pigs because it would be too expensive to implement and has the potential to disrupt the pig value chain when large numbers of market-ready pigs are taken off the farms for research purposes. To account for the poor serological diagnostic performance, our reported prevalence estimates for PCC have been adjusted to reflect the test's sensitivity and specificity.

Although we recognize the reduced specificity of the ELISA test used in this study, the fact that cases of PCC have been confirmed shows that there is the existence of active taeniasis cases which provides substantial evidence to support a case for investing in control interventions like mass deworming programs in humans. In an observational study, misclassification of the outcome may occur, but there was no reason to believe that this misclassification was differential (i.e., different at different levels of predictors) so it is almost certain that this misclassification would have biased observed relationships toward the null (i.e., an underestimation of the effects of observed risk factors) and a corresponding reduction in the power of the study [26]. Consequently, our estimates of the effects of risk factors are conservative. We suggest that future large-scale studies can use AgELISA with an adjustment of the sample size to account for the loss of power attributable to misclassification bias, along with limited dissections as we have done.

## Conclusion

This study confirms that the area is a hyperendemic foci for *T. solium* infections in Uganda, as predicted by the geospatial models. Although efforts to control the parasite have been stimulated through successive roadmaps to control Neglected Tropical Diseases (NTD), hyperendemic areas with ongoing *Taenia solium* transmission remain, such as the one identified in Northern Uganda. The wide distribution of cases across the area warrants the initiation of region-wide control interventions using the One Health approach if the targets laid out in the 2021–2030 WHO roadmap for control of NTDs are to be achieved. For example, an intervention focusing on the treatment of both humans and pigs with an addition of provision of health education to improve hygiene and husbandry practices.

## Supporting information

**S1 Table. Variables collected for the Analysis.**
(DOCX)

**S2 Table. Sources and data types of environmental variables.**
(DOCX)

**S3 Table. Summary of the number of villages, households and pigs sampled.**
(DOCX)

**S4 Table. Key pig husbandry practices.**
(DOCX)

**S5 Table. Table showing the unconditional association of variables with pig seropositivity.**
(DOCX)

**S6 Table. Results of the univariable analysis of environmental factors.**
(DOCX)

**S1 Text. Results of K-means clustering analysis.**
(DOCX)

**S1 Fig. Village level Random effects plot.**
(TIF)

**S2 Fig. Normality plot for village level random effects.**
(TIF)

## Acknowledgments

The authors would like to acknowledge all the participating pig farmers for their willingness to engage with this study. We would also like to thank Samuel Kidega, Christine Lamwaka, Tabu Richard and Otika Muzamil for their support during data collection and conducting carcass dissection. All the meat inspectors, traders, private veterinarians, and Chairperson LC1 in the study areas are also appreciated for welcoming, guiding and supporting the team during data collection. The team at the CDL at Makerere University led by Dr. Wampande Edward and Kayaga Edrine are highly appreciated for their support in sample analysis. Special appreciation is also due to the Administration team at ILRI-Uganda for the administrative and logistical support during fieldwork.

## Author contributions

**Conceptualization:** Nicholas Ngwili, Salaviriuse Ahimbisibwe, Stephen Bole, Emily Ouma, Ian Dohoo, Lian F. Thomas.

**Data curation:** Nicholas Ngwili, Salaviriuse Ahimbisibwe, Stephen Bole, Lian F. Thomas.

**Formal analysis:** Nicholas Ngwili, Salaviriuse Ahimbisibwe, Max Korir, Stephen Bole, Ian Dohoo, Lian F. Thomas.

**Funding acquisition:** Lian F. Thomas.

**Investigation:** Nicholas Ngwili, Salaviriuse Ahimbisibwe, Stephen Bole, Alfred Kinyera, Stanely Vusso Avudraga, Robert Saverio Okeny, Robert Okwera Kenny, Lian F. Thomas.

**Methodology:** Nicholas Ngwili, Salaviriuse Ahimbisibwe, Stephen Bole, Ian Dohoo, Lian F. Thomas.

**Project administration:** Nicholas Ngwili, Salaviriuse Ahimbisibwe, Stephen Bole, Lian F. Thomas.

**Supervision:** Clovice Kankya, Emily Ouma.

**Validation:** Nicholas Ngwili, Salaviriuse Ahimbisibwe, Max Korir, Stephen Bole, Ian Dohoo.

**Visualization:** Nicholas Ngwili, Max Korir, Stephen Bole, Ian Dohoo.

**Writing – original draft:** Nicholas Ngwili, Salaviriuse Ahimbisibwe, Stephen Bole, Lian F. Thomas.

**Writing – review & editing:** Nicholas Ngwili, Salaviriuse Ahimbisibwe, Max Korir, Stephen Bole, Clovice Kankya, Alfred Kinyera, Stanely Vusso Avudraga, Robert Saverio Okeny, Robert Kenny Okwera, Emily Ouma, Ian Dohoo, Lian F. Thomas.

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
