## [Decision Letter · Decision Letter 0]

21 May 2025

Response to Reviewers
Revised Manuscript with Track Changes
Manuscript

Shaden Kamhawi

co-Editor-in-Chief

Paul Brindley

co-Editor-in-Chief

**Journal Requirements:**

- ® on page: 10 and 11.

Potential Copyright Issues:

- Figure 2. Please (a) provide a direct link to the base layer of the map (i.e., the country or region border shape) and ensure this is also included in the figure legend; and (b) provide a link to the terms of use / license information for the base layer image or shapefile. We cannot publish proprietary or copyrighted maps (e.g. Google Maps, Mapquest) and the terms of use for your map base layer must be compatible with our CC BY 4.0 license.

6) When completing the data availability statement of the submission form, you indicated that you will make your data available on acceptance. We strongly recommend all authors decide on a data sharing plan before acceptance, as the process can be lengthy and hold up publication timelines. Please note that, though access restrictions are acceptable now, your entire data will need to be made freely accessible if your manuscript is accepted for publication. This policy applies to all data except where public deposition would breach compliance with the protocol approved by your research ethics board. If you are unable to adhere to our open data policy, please kindly revise your statement to explain your reasoning and we will seek the editor's input on an exemption. Please be assured that, once you have provided your new statement, the assessment of your exemption will not hold up the peer review process.

7) Please amend your detailed Financial Disclosure statement. This is published with the article. It must therefore be completed in full sentences and contain the exact wording you wish to be published. Please ensure that the funders and grant numbers match between the Financial Disclosure field and the Funding Information tab in your submission form. Note that the funders must be provided in the same order in both places as well.

**Reviewers' comments:**

**Key Review Criteria Required for Acceptance?**

**Methods:**

-Are the objectives of the study clearly articulated with a clear testable hypothesis stated?

-Is the study design appropriate to address the stated objectives?

-Is the population clearly described and appropriate for the hypothesis being tested?

-Is the sample size sufficient to ensure adequate power to address the hypothesis being tested?

-Were correct statistical analysis used to support conclusions?

-Are there concerns about ethical or regulatory requirements being met?

Reviewer #1: The global distribution of Taenia solium is widespread, and taeniasis caused by T. solium is a significant foodborne parasitic disease that has garnered increasing attention in recent years. Conducting epidemiological surveys on taeniasis and cysticercosis in highly endemic regions is of great importance. The author’s investigation on porcine cysticercosis in northern Uganda could provide valuable evidence for understanding the prevalence of T. solium and verify the accuracy of current epidemiological risk assessments in the region. However, the study has several limitations, outlined as follows:

The incidence of taeniasis and cysticercosis exhibits a parallel and concurrent relationship. Therefore, investigations into human infections with adult T. solium (e.g., fecal sampling for egg detection, areca nut-pumpkin seed deworming, and human serological testing) are equally critical. Additionally, the occupational status of participants (e.g., involvement in slaughtering) and dietary habits (e.g., consumption of raw pork or vegetables) are strongly correlated with cysticercosis incidence and should be included in the survey.

To enhance the representativeness of the findings, the methodology should include not only population density but also details such as the surveyed area’s size, village distribution, and household density.

Reviewer #2: The methodological approach is generally respected, with the exception of a few specific points, notably on the sampling methodology, which requires more clarification and adjustment according to the results obtained.

It is important to note that the sample size is large enough to guarantee sufficient statistical power for the study.

The writing is clear and accessible

**Results**

-Does the analysis presented match the analysis plan?

-Are the results clearly and completely presented?

-Are the figures (Tables, Images) of sufficient quality for clarity?

Reviewer #1: The manuscript also contains several issues The inappropriate presentation of these results may undermine the entire article's quality.:

舄Table 1舄: The unit for "Population density" is unspecified. What do the symbols "HH," "HH*," and "HH" denote?

舄Line 217舄: The "Data analysis" section lacks a description of statistical software used.

舄Table 2舄: Are the categorized data (sex, age, reproduction) for pigs presented as proportions or absolute numbers? If proportions, include "%." For example, 270 × 0.433 = 116.91 (non-integer pig counts are illogical). In the "Breed" category, the first group’s data (73.3 + 36.7 = 110) exceed 100%.

舄Table 3舄: Similarly, numerical values lack clarification (proportions or counts?).

舄References舄: Many citations are listed; ensure the referencing style complies with journal guidelines.

Reviewer #2: The result aligns with the analysis plan, except for a few adjustments or modifications that need to be made in the multivariate analysis.

Table 3 requires an introductory text for interpretation.

The table numbers need to be revised

**Conclusions**

-Are the conclusions supported by the data presented?

-Are the limitations of analysis clearly described?

-Do the authors discuss how these data can be helpful to advance our understanding of the topic under study?

-Is public health relevance addressed?

Reviewer #1: (No Response)

Reviewer #2: The subject is original and of great significance form both scientifically and operationally. A concrete example in the discussion part on the proposed One Health approach adapted to their findings would be nice.

**Editorial and Data Presentation Modifications?**

Reviewer #1: (No Response)

Reviewer #2: Minor revision.

**Summary and General Comments**

Reviewer #1: (No Response)

Reviewer #2: The study is grounded in a clearly defined research question and addresses a methodological gap in assessing the prevalence of porcine cysticercosis, which serves as a proxy for the evaluation and control of neurocysticercosis.

I have only a few comments, attached in this review, to help improve the manuscript prior to publication

PLOS authors have the option to publish the peer review history of their article (what does this mean? ). If published, this will include your full peer review and any attached files.

**Do you want your identity to be public for this peer review?** For information about this choice, including consent withdrawal, please see our Privacy Policy .

Reviewer #1: No

Reviewer #2: No

**Figure resubmission:****Reproducibility:** To enhance the reproducibility of your results, we recommend that authors of applicable studies deposit laboratory protocols in protocols.io, where a protocol can be assigned its own identifier (DOI) such that it can be cited independently in the future. Additionally, PLOS ONE offers an option to publish peer-reviewed clinical study protocols. Read more information on sharing protocols at https://plos.org/protocols?utm_medium=editorial-email&utm_source=authorletters&utm_campaign=protocols

---

## [Decision Letter · Decision Letter 1]

2 Jul 2025

Dear Dr. Thomas,

We are pleased to inform you that your manuscript 'Confirmation of a hyperendemic focus of porcine cysticercosis in Northern Uganda: Prevalence and risk factor analysis' has been provisionally accepted for publication in PLOS Neglected Tropical Diseases.

Best regards,

Francesca Tamarozzi

Section Editor

Gabriel Rinaldi

Section Editor

Shaden Kamhawi

co-Editor-in-Chief

Paul Brindley

co-Editor-in-Chief

Reviewer's Responses to Questions

**Key Review Criteria Required for Acceptance?**

**Methods**

-Are the objectives of the study clearly articulated with a clear testable hypothesis stated?

-Is the study design appropriate to address the stated objectives?

-Is the population clearly described and appropriate for the hypothesis being tested?

-Is the sample size sufficient to ensure adequate power to address the hypothesis being tested?

-Were correct statistical analysis used to support conclusions?

-Are there concerns about ethical or regulatory requirements being met?

Reviewer #1: Yes, all the specified aspects are fully addressed in this study.

Reviewer #2: (No Response)

**Results**

-Does the analysis presented match the analysis plan?

-Are the results clearly and completely presented?

-Are the figures (Tables, Images) of sufficient quality for clarity?

Reviewer #1: Two minor points for the author to confirm

Line 83�”Humans are the definitive hosts for the mature tapeworm as well as the aberrant hosts in cases of human cysticercosis”�I think that in this sentence, "aberrant hosts" was not very appropriate and should be changed to "intermediate hosts".

Line 303: “Other predictors were retained if P<.05 or P<0.15 for environmental variables”, 舄Change "P<.05" to "P<0.05", please confirm.

Reviewer #2: (No Response)

**Conclusions**

-Are the conclusions supported by the data presented?

-Are the limitations of analysis clearly described?

-Do the authors discuss how these data can be helpful to advance our understanding of the topic under study?

-Is public health relevance addressed?

Reviewer #1: Yes, all the specified aspects are fully addressed in this study.

Reviewer #2: (No Response)

**Editorial and Data Presentation Modifications?**

Reviewer #1: I recommend acceptance of the manuscript.

Reviewer #2: Accept

**Summary and General Comments**

Reviewer #1: The global distribution of Taenia solium is widespread, and taeniasis caused by T. solium is a significant foodborne parasitic disease that has garnered increasing attention in recent years. Conducting epidemiological surveys on taeniasis and cysticercosis in highly endemic regions is of great importance. The author’s investigation on porcine cysticercosis in northern Uganda could provide valuable evidence for understanding the prevalence of T. solium and verify the accuracy of current epidemiological risk assessments in the region.

Reviewer #2: Feed back OK

PLOS authors have the option to publish the peer review history of their article (what does this mean? ). If published, this will include your full peer review and any attached files.

**Do you want your identity to be public for this peer review?** For information about this choice, including consent withdrawal, please see our Privacy Policy .

Reviewer #1: No

Reviewer #2: No

---

## [Editor Report · Acceptance letter]

Dear Dr. Thomas,

We are delighted to inform you that your manuscript, " 

Confirmation of a hyperendemic focus of porcine cysticercosis in Northern Uganda: Prevalence and risk factor analysis," has been formally accepted for publication in PLOS Neglected Tropical Diseases.

Best regards,

Shaden Kamhawi

co-Editor-in-Chief

Paul Brindley

co-Editor-in-Chief
